# The Prognostic Value of Platelet–Lymphocyte Ratio, Neutrophil–Lymphocyte Ratio, and Monocyte–Lymphocyte Ratio in Head and Neck Squamous Cell Carcinoma (HNSCC)—A Retrospective Single Center Study and a Literature Review

**DOI:** 10.3390/diagnostics13223396

**Published:** 2023-11-07

**Authors:** Camil Ciprian Mireștean, Mihai Cosmin Stan, Roxana Irina Iancu, Dragoș Petru Teodor Iancu, Florinel Bădulescu

**Affiliations:** 1Department of Medical Oncology and Radiotherapy, University of Medicine and Pharmacy Craiova, 200349 Craiova, Romania; mc3313@yahoo.com (C.C.M.); fbadulescu2001@yahoo.com (F.B.); 2Department of Surgery, Railways Clinical Hospital, 700506 Iasi, Romania; 3Department of Medical Oncology, Vâlcea County Emergency Hospital, 200300 Râmnicu Vâlcea, Romania; 4Faculty of Dental Medicine, Oral Pathology Department, “Grigore T. Popa” University of Medicine and Pharmacy, 700115 Iasi, Romania; roxana.iancu@umfiasi.ro; 5Department of Clinical Laboratory, “Saint Spiridon” County Hospital, 700111 Iasi, Romania; 6Oncology and Radiotherapy Department, Faculty of Medicine, “Gr. T. Popa” University of Medicine and Pharmacy, 700115 Iași, Romania; 7Department of Radiation Oncology, Regional Institute of Oncology, 700483 Iași, Romania

**Keywords:** head and neck cancers, HNC, HNSCC, NLR, PLR, MLR, biomarkers, prognostic

## Abstract

Introduction: Neutrophil-to-lymphocyte ratio (NLR), lymphocyte-to-monocyte ratio (LMR), and pallets-to-lymphocyte ratio (PLR) are currently validated as cheap and accessible biomarkers in different types of solid tumors, including head and neck cancers (HNC). The purpose of the study: To evaluate the possible purposes and biomarker value of NLR, PLR, and MLR recorded pre-treatment (radiotherapy/chemotherapy) in HNC. Materials and methods: From 190 patients with HNC included in the oncology records in the oncology outpatient clinic of the Craiova County Emergency Hospital (from January 2002 to December 2022), 39 cases met the inclusion criteria (squamous cell carcinoma and the possibility to calculate the pre-treatment (chemotherapy/radiotherapy) value of NLR, PLR, and MLR. Overall survival (OS) values were correlated with NLR, PLR, and MLR. Results: The median values for NLR, PLR, and MLR were 6.15 (1.24–69), 200.79 (61.3–1775.0), and 0.53 (0.12–5.5), respectively. In the study, the mean values for NLR, PLR, and MLR of 2.88, 142.97, and 0.36, respectively, were obtained. The median OS in the study group was 11 months (1–120). Although a negative Pearson’s correlation was present, the relationship between the variables was only weak, with values of R = 0.07, *p* = 0.67, R = 0.02, *p* = 0.31, and R = 0.07, *p* = 0.62 being related to NLR, PLR, and MLR, respectively, in correlation with OS. The median values of NLR, PLR, and MLR were calculated (1.53, 90.32, and 0.18, respectively) for the HNC cases with pre-treatment values of NLR < 2 and for the HNC cases with NLR values ≥ 6 (23.5, 232.78, and 0.79, respectively). The median OS for cases with NLR < 2 and NLR ≥ 6 were 17.4 and 13 months, respectively. Conclusions: The comparative analysis of the data highlights a benefit to OS for cases low values of NLR. The role of not only borderline NLR values (between 2 and 6) as a prognostic marker in HNSCC but also the inclusion of PLR and MLR in a prognostic score must also be defined in the future. Prospective studies with more uniformly selected inclusion criteria could demonstrate the value of pre-treatment NLR, PLR, and MLR for treatment stratification through the intensification or de-escalation of non-surgical curative treatment in HNSCC.

## 1. Introduction

The role and value of peripheral blood biomarkers have been demonstrated in several types of cancer, especially in recent years. Neutrophil-to-lymphocyte ratio (NLR), lymphocyte-to-monocyte ratio (LMR), and pallets-to-lymphocyte ratio (PLR) have been investigated independently or in association and are evaluated markers of inflammation, with increased values being correlated with unfavorable prognosis. NLR, one of the most frequently evaluated biomarkers, is correlated with two factors: acute and chronic inflammation indicated by neutrophil count and adaptive immunity correlated with the number of lymphocytes. PLR can be associated with the release of cytokines and chemokines, and MLR reflects both inflammation and the risk of cancer progression, with an alteration of this ratio involving the inability of the host to block tumor proliferation [1,2,3]. One of the major advantages of using NLR, PLR, and MLR as biomarkers is the simple, inexpensive evaluation methods involved, the patient’s discomfort being minimal, and only peripheral blood samples being required. In contrast, the modern methods of immunohistochemistry and polymerase chain reaction (PCR) are mentioned, which involve taking samples by means of biopsy [4].

## 2. NLR, PLR, and MLR—Growing Interest in Human Papilloma Virus (HPV)-Related HNC

The heterogeneity of the response to aggressive multimodal treatment, including surgery, radiotherapy, chemotherapy, targeted molecular therapy, and immunotherapy justifies the efforts to identify prognostic and predictive biomarkers in head and neck cancers (HNC). Even if the subtype of the disease associated with HPV infection is related to a superior response to treatment and a favorable prognosis, a stratification of the treatment to not only limit the toxicities associated with the treatment but also increase the response rate is still the subject of clinical studies. Understanding not only the physiopathological mechanisms involved, the role of inflammation, and the host’s immune response in initiating cancer progression and metastasis but also in modulating the response to treatment, explaining a favorable response due to the abundance of immune cells in HPV-related HNC, are arguments that explain the growing interest in NLR, PLR, and MLR as possible biomarkers in HNC [5]. NLR is the most frequently reported biomarker, and the meta-analysis of Rodrigo et al., which includes studies related to oropharyngeal cancer, identifies NLR, more precisely the increased values of this ratio, as a negative prognostic factor. Even though, regardless of HPV status, high pre-treatment NLR values were associated with an unfavorable prognosis in oro-pharyngeal cancer, the correlation of NLR with prognosis was more significant in cases of HPV-related oropharyngeal cancer [6].

The aim of the study was to evaluate the possible biomarker value of NLR, PLR, and MLR recorded pre-treatment for prognostic purposes in HNC.

## 3. NLR, PLR, and MLR—From Prognostic Markers to Future Orchestrators of an HNC Multimodal Approach?

The rapid development in recent years of genetic and molecular analysis has made it possible to identify specific molecular changes that could guide oncological treatments in HNC. However, the absence of clear correlations between molecular and genetic features and a prognosis makes studies on large samples and uniform characterizations of tumor specimens necessary. Excision repair cross-complementing group 1 (ERCC1), a deoxyribonucleic acid (DNA) repair enzyme involved in Cisplatin resistance and HPV status, is the most promising biomarker in HNC. However, the generation of combined markers including not only imaging/radiomics features and genetic and molecular mutation but also peripheral blood biomarkers could predict not only prognosis but also the pattern of evolution in order to stratify the treatment escalation and elective management of neck lymph nodes [7]. Three reports including the value of neutrophils were analyzed in a systematic review of 49 relevant studies that included HNC patients. Even if obvious heterogeneity was observed between the studies, NLR, PLR, and MLR were strongly correlated with prognosis. Kumarasamy et al. consider that these biomarkers could be introduced into clinical practice for therapeutic decisions [8]. A study conducted in the Department of Otolaryngology, Head and Neck Surgery of Beijing Tongren Hospital included 50 laryngeal cancer patients and 40 healthy subjects as a control group and evaluated NLR, lymphocyte-to-monocyte ratio (LMR), and monocyte-to-white blood cell ratio (MWR) as prognostic markers. The values calculated were higher for NLR, PLR, and MWR and were lower for LMR in the group of patients with cancer than in the control group. Increased values of NLR, PLR, and MWR and lower values of LMR were associated with a reduced 5-year OS; this positive correlation for NLR, PLR, and MWR and negative correlation for LMR was also identified in the case of a comparative assessment of survivors and deceased patients. Age, alcohol consumption, and smoking as well as TNM staging were correlated with these biomarkers, it being considered that their association could confer superior performance [9]. A significant increase in NLR and PLR with the degree of histological tumor differentiation and TNM staging was observed in a study that included 170 HNC cases and 80 cases as the control group. The increased values of NLR and PLR were also higher in the HNC cases compared to the values obtained in healthy patients, but they were also correlated with an unfavorable prognosis. The authors recommend confirmation of the results on larger patient groups and assessment of these markers’ correlations with the treatment response [10,11].

## 4. NLR, PLR, and MLR—Optimal Cut-Off Value or Reference Interval—A Unsolved Dilemma

An NLR value ≥ 3.6 was considered as the cut-off value in a group of 180 patients from Taiwan diagnosed with advanced stages (stage III and IV) of nasopharyngeal cancer. NLRs higher than the maintained value are associated with reduced PFS, DFS, and OS [12].

Brewczyński and colleagues proposed the evaluation of NLR, PLR, MLR, and the SII systemic inflammation index before and after radiotherapy/chemotherapy in patients diagnosed with oropharyngeal cancers related and unrelated to HPV infection, with the researchers trying to identify a correlation of these parameters with DFS and OS. In the case of HPV-positive patients, a correlation with reduced OS was identified not only for increased pretreatment white blood cells (WBCs) (>8.33/mm^3^) but also with values for NLR > 2.13 and SII > 448.60. Values for NLR > 2.29 and SII > 462.58 were strongly associated with reduced DFS. In the case of patients with HPV-unrelated disease, these correlations were not identified [13].

A meta-analysis that included 24 articles and an independent set of patients (*n* = 540) evaluated the values and ranges of values of NLR that were correlated with OS and DFS. The study did not identify a significant cut-off for OS and DFS in the range of NLR > 2.2 and <6, with significant differences being identified for values of NLR < 2.2 and >6; higher values for NLR are associated with an unfavorable prognosis [14]. Analyzing 28 cohorts including 6847 head and neck squamous cell carcinoma (HNSCC) patients, a systematic review and meta-analysis by Yang et al. concluded that increased values of pre-treatment NLR are not only associated with DFS, PFS, and OS but also cancer-specific survival, without a strong correlation [15].

Evaluating the prognostic value of NLR, MLR, PLR, alkaline phosphatase (ALP), and lactate dehydrogenase (LDH) pre-operatively and post-operatively, a study that included 361 cases of squamous laryngeal cancer identified the value of NLR, PLR, and MLR in both perioperative settings as prognostic markers. ALP and LDH did not demonstrate predictive power in all situations. The authors propose preoperative NLR and postoperative MLR as independent markers of OS and PFS in laryngeal cancer [16].

The heterogeneity in the cut-off values reported by different authors justifies the initiation of studies with the aim of identifying this value with a prognostic role. Using the maximum concordance index (C-index) method and internal validation via the bootstrapping method, the NLR cut-off was identified in numerous non-metastatic nasopharyngeal cancer patients treated with intensity-modulated radiotherapy (IMRT) with curative intention. The study included 463 patients and the follow-up period was 70.8 months. The value of NLR = 3 was associated with the highest C index (0.548). An NLR > 3 was considered an independent prognostic factor, with it being associated with reduced survival. The authors recommend the introduction of this prognostic factor in the evaluation of naso-pharyngeal cancer [17].

The prognostic value of NLR and PLR in patients with HNSCC treated with definitive or adjuvant chemo-radiotherapy using complete blood counts (CBCs) recorded 10 days before the start of treatment were evaluated in a study that included 186 patients in relation to OS, locoregional recurrence-free survival (LRFS), DFS, and acute toxicity. Most cases (45%) were cancers of the oropharynx, followed by cancers of the oral cavity, hypopharynx, and larynx in proportions of 28%, 14%, and 13% respectively. The study identified a relationship between NLR and OS and between LTFS and DFS. Acute grade ≥ 2 toxicity was not correlated with any of these markers. PLR was also not associated with outcomes or toxicity. The authors considered that NLR could be used as an independent predictive biomarker of mortality in patients with HNSCC treated with chemo-radiotherapy [18]. Takenada et al. analyzed 19 studies that included 3770 patients and demonstrated the prognostic value of NLR in HNSCC. The results highlighted in all studies a correlation of the NLR values higher than the cut-off with poorer disease-specific survival and OS. A meta-analysis proposed by the same authors that included 9 studies that enrolled 2327 patients also demonstrated the prognostic value of PLR in HNSCC. In this case, higher PLR values were associated with an unfavorable prognosis. Another study by a team from an otorhinolaryngology department in Osaka, Japan, identified NLR as a predictor of response to immune checkpoint inhibitor (ICI) treatment in HNSCC [19,20,21]. A 1.78 combined and site-specific hazard ratio for OS was identified by Mascarella et al. in a meta-analysis including 24 studies and 6479 patients for cases with higher NLR values (ranging between 2.04–5). The highest hazard ratio for OS was 2.36, with this value being associated with hypopharyngeal cancer [22]. PLR and NLR were recorded in the first 4 weeks of treatment in a group of 273 HNSCC patients treated at McGill University Health Center in a time interval of 11 years. The study evaluated PLR and NLR in relation to recurrence and mortality rates. Increased mortality (43%) and a more advanced T stage were associated with PLR > 170 and NLR ≤ 3.0. NLR values above 4.2 were associated with a higher risk of recurrence. The study identifies the association between NLR and PLR at least as precise as TNM staging in predicting survival [23].

Using the AUROC peaks method, the optimal threshold for 5-year survival and values for NLR and PLR were calculated in HNSCC patients treated with surgery. Cases without R0 resection and patients with chronic inflammatory diseases were excluded from this study. Cut-off values of 113 and 2.8 for PLR and NLR, respectively, were identified. It should be noted that there were different proportions of patients with PLR values higher than the threshold in the case of hypopharyngeal tumors (71.7%) and lower than the threshold in the case of oropharyngeal tumors (25.0%). Also, mortality at 5 years in the case of tumors with PLR values lower than the threshold was 24.6% vs. 46.4% in the case of tumors with PLR ≥ 107. And for NLR, the same indicator was 32.3% vs. 56.5% for NLR values < 3.9 and NLR values ≥ 3.9, respectively. For PLR, the correlation was also observed in the case of DFS and cancer-specific mortality. NLR was correlated only with mortality [24]. The prognostic values (including OS and DFS) of pre-treatment lymphocyte-to-monocyte ratio (LMR) in HNSCC patients were evaluated in a systematic review and a meta-analysis that included 4260 cases from 7 cohorts. The increased value of LMR was associated with a favorable prognosis. The authors mention the need for carreful evaluation of the results, with it being a retrospective study [25]. Changes in the dynamics of LMR during radiotherapy for HNC were evaluated in relation to OS and metastasis-free survival in a group of 1431 patients. The follow-up period was 9 years, and during this period 44.4% of the patients died and 16.8% developed distant metastases. Higher delta-MLR variation at 2 weeks was associated with OS and metastasis-free survival at 5 years of 59% and 80% rates, respectively. In the case of lower delta-MLR, the same variables were associated with 73% and 87%, respectively. Delta-MLR was identified as an independent prognostic factor in HNC patients treated with radiotherapy. The authors recommend the use of this biomarker, with it being cheap and accessible [26].

An analysis that included 215 cases that fulfilled the study criteria of primary adenocarcinoma and carcinoma and advanced stage sino-nasal cancers, highlighted a correlation between the pre-treatment values of NLR and PLR with OS and DFS. Cases with higher values of the two markers had shorter OS and DFS. NLR < 2.6 and PLR < 156.9 were associated with reduced risk of recurrence [27]. A complex score involving fibrinogen value and NLR was proposed for prognostic purposes in advanced hypopharyngeal carcinoma. All 111 patients in the study were treated with radiotherapy, bio-radiotherapy, or chemo-radiotherapy. Three score values were used to divide the patients into groups, and subsequently, the prognostic value of the scores was analyzed. Fibrinogen ≥ 341 mg/dL and NLR ≥ 3.59, i.e., F-SCOR = 2, were identified as independent predictors of OS and PFS. In the case of patients with F-NLR score = 2, OS and PFS were significantly lower than in the group with F-NLR score = 0 (fibrinogen < 341 mg/dL and NLR < 3.59) [28]. The studies considered significant for identifying not only the predictive power for treatment response but also the prognostic value of NLR, PLR, and MLR are summarized in Table 1 [9,10,11,12,13,14,15,16,17,18,19,20,21,22,23,24,25,26,27,28,29,30].

## 5. Materials and Methods

The study lot included 190 patients with HNC identified from the oncology records at the oncology outpatient clinic of the Craiova County Emergency Hospital starting from January 2002 until December 2022, including those that had died. Among them, only 40 patients multimodally treated in the oncology clinic of the county hospital were included in the study, in conformity with the inclusion and exclusion criteria. Squamous cell carcinoma histopathology and the possibility to calculate not only the pre-chemotherapy value of NLR, PLR, and MLR but also OS were the main inclusion criteria. Cases that benefited from surgical interventions for diagnostic, curative, and palliative purposes were accepted. Synchronous or metachronous malignancies or autoimmune diseases treated with immunosuppressive medication were also included in the exclusion criteria. Hypertension and type 2 diabetes mellitus were not considered in the exclusion criteria. Other systemic oncological treatments and radiotherapy before the evaluation of these markers were considered in the exclusion criteria. A complete blood test was recorded on the day of admission to the oncology department, with it being included in the evaluation of chemotherapy regimen administration eligibility.

All patients received chemotherapy with curative or palliative monotherapy based on platinum salts or regimens of platinum doublet or triplet combinations including taxanes, 5-flurouracil, or oral analog capecitabine, with the number of chemotherapy cycles varying between 1 and 12. The choice of the chemotherapy regimen was the physician’s choice or also influenced by the temporarily limited accessibility to certain chemotherapeutic agents in Romania. It should be mentioned that, with this being a retrospective study, in the cases that required surgery, the referral to the oncology clinics followed the surgical procedure. For this reason, the NLR, PLR, and MLR values were recorded before chemotherapy/radiotherapy and not before treatment in all cases. Pearson’s correlation of NLR, PLR, and MLR with overall survival (OS) values was also analyzed.

### Study Limits and Possible Excluded Confounding Factors

The small number of patients included in the study is explained by the fact that a large number of cases that were enrolled in the oncology record of the department had initiated chemotherapy or radiotherapy in other departments, and consequently, the evaluation of the nadir value (pre-chemo-radiotherapy) was not accessible.

A complete blood test was repeated before each cycle of chemotherapy, but the dynamic assessment of biomarkers was not proposed because the variable time interval between chemotherapy courses and the differences between the radiotherapy regimens delivered sequentially or concurrently were also considered confounding factors. We would also like to mention that due to the long period of inclusion in the study, there was variability in the technique and radiotherapy doses, from a Rokus M40 cobalt machine without imaging guidance to a three-dimensional conformal technique based on a LINAC machine. The variability in the clinical or imaging assessment of progression and the limitations due to technical reasons for the computer tomography (CT) follow-up persuaded us not to include DFS and PFS in the analysis.

Evaluating the data from the meta-analysis by Cho and colleagues, we proposed in the analysis a comparison of the data for lower and higher values of NLR (<2 and ≥6, respectively) [14].

## 6. Results

Of the 40 patients included in the study, 39 (97.5%) were diagnosed with head and neck squamous cell carcinoma (HNSCC). One patient was diagnosed with pleomorphic sarcoma. After the initial analysis, this case was excluded from the study; only cases of HNSCC were included. The average age at the onset of disease was 64.84 years (48–86 years). Hypertension and type 2 diabetes mellitus were reported in eight and two cases, respectively. In the vast majority of cases, 35 out of the 39 (89.7%) patients included in the study were male and 4 out of 39 (10.3%) were female. All cases were diagnosed with locally advanced or metastatic stages (III and IV). In total, 3 cases were diagnosed with stage IVB (two loco-regional recurrences and one lung metastasis), 4 cases were diagnosed with stage IVA, and 32 cases were diagnosed with stage III. The group of patients included various anatomical sites of disease: oropharynx—11 cases (28.2%), larynx—7 cases (17.94%), oral cavity—13 cases (33.33%), nasopharynx—cases (2.55%), unknown primary—2 cases (5.1%), parathyroid—1 case (2.55%), and sinonasal—1 case (2.55%) (Table 1). A heatmap was chosen to intuitively represent the correlation of NLR, MLR, and PLR with OS. A color scale from red (chosen for low values to represent the worst OS) to orange, yellow to green (for the highest values) highlights in most cases, especially in those with extreme values, the proportionality of the analysis variables with the prognosis (Table 2). All cases were histo-pathologically proven head and neck squamous cell carcinoma (HNSCC). The mean nadir values for neutrophils, lymphocytes, platelets, and monocytes absolute count were: 6630 (560–18,000), 1815 (200–4240), 268,000 (75,000–622,000), and 670 (100–2000), respectively. The mean values for NLR, PLR, and MLR were 6.22 (1.24–69), 203.17 (61.3–1775.0), and 0.53 (0.12–5.5), respectively. In the study, the median values for NLR, PLR, and MLR of 2.88, 142.97, and 0.36, respectively, were obtained. For cases aged <70 years, the mean nadir values for neutrophils, lymphocytes, platelets, and monocytes absolute count were: 6115 (560–17,210), 1723 (410–2730), 271,500 (82,550–49,500) and 658 (190–2000), respectively. The mean values for NLR, PLR, and MLR were 3.56 (1.37–8.61), 166.71 (117.50–214.29), and 0.53 (0.14–0.42), respectively. The nadir values for neutrophils, lymphocytes, platelets, and monocytes absolute count were: 6823 (1830–179,700), 1849 (200–4240), 270,366 (75,000–622,000), and 665 (100–1550), respectively. The mean values for NLR, PLR, and MLR were 6.91 (1.24–69), 212.58 (61.63–1775), and 0.56 (0.12–5.5), respectively, and they were identified in cases of HNSCC aged ≥70. For the group of patients aged <70 years, the mean age in the study group was 67.7 (48–69) years, with the median OS being 12 months. Patients aged ≥70 years had a mean age of 77 (70–86) years, and the median OS was 85 months. For cases of oral cavity HNSCC, the nadir values for neutrophils, lymphocytes, platelets, and monocytes absolute count were: 7317 (1830–17,210), 1795 (400–4240), 313,120 (82,560–622,000) and 640 (100–2000), respectively. The mean values for NLR, PLR, and MLR were 6.19 (1.39–31.85), 206.19 (88.44–392.75), and 0.42 (0.14–10), respectively. The mean age in the group of oropharyngeal cancer cases was 65.15 (48–86) years, with the median OS being 11 months. For the second subtype of HNSCC incidence in the study group, oropharyngeal cancer was associated with nadir values for neutrophils, lymphocytes, platelets, and monocytes absolute counts of 5866 (275–12,660), 1699 (580–3180), 196,780 (100,500–335,000), and 587 (290–1430). The mean values for NLR, PLR, and MLR were 4.36 (1.48–15.50), 125.2 (70.43–174.14), and 0.39 (0.13–0.76), respectively. The mean age in the group of oropharyngeal cancer cases was 62.4 (54–67) years, with the median OS being 15 months. In cases of laryngeal cancer, the nadir values for neutrophils, lymphocytes, platelets, and monocytes absolute count were: 6653 (2600–12,660), 1935 (580–4010), 263,488 (75,000–502,000), and 664 (290–1170). The mean values for NLR, PLR, and MLR were 4.05 (1.25–9.22), 149.63 (70.06–247.6), and 0.41 (0.14–0.85), respectively. In the group of laryngeal cancer cases, the median age was 64.6 (55–76) years, with a median OS of 15 months. The free online platform https://www.socscistatistics.com/ (accessed on 14 September 2023) was used for the statistical analysis. In the study, the mean values for NLR, PLR, and MLR of 2.88, 142.97, and 0.36, respectively, were obtained. The median OS in the study group was 11 months (1–120). Although a negative Pearson’s correlation was observed, the relationship between the variables was only weak, with values of R = 0.07, *p* = 0.67; R = 0.02, *p* = 0.31; and R = 0.07, *p* = 0.62 being related to NLR, PLR, and MLR, respectively, and in correlation with overall survival (OS). The mean values of NLR, PLR, and MLR were calculated (1.53, 90.32, and 0.18, respectively) for HNC cases with pre-treatment values of NLR < 2 and for HNC cases with NLR values ≥ 6 (23.5, 232.78, and 0.79, respectively). The characteristics of the patients and the heatmap including NLR, PLR, MLR, and OS (in months) for 39 HNSCC cases are also included in Table 2 and Table 3, respectively. Figure 1 compares the mean nadir values of NLR for patients with nadir NLR values between 0 and 2 and higher than 6. Figure 2 and Figure 3 compare the mean nadir values of PLR and MLR, respectively, for cases with nadir NLR values between 0 and 2 and higher than 6. It can be observed that in both cases, the higher median values of NLR are also associated with an increase in the mean values of PLR and MLR. The median OS for cases with NLR < 2 and NLR ≥ 6 were 17.4 and 13 months, respectively (Figure 4). Figure 5, Figure 6 and Figure 7 compare the nadir values of NLR, PLR, and MLR for the HNSCC subtypes (laryngeal, oropharyngeal, and oral cavity). Oral cavity HNSCC is associated with the highest nadir values of all of these markers. The laryngeal cancer cases had the lowest nadir mean value for NLR, and oropharyngeal cancer cases were related to the lowest nadir mean values of PLR and MLR.

## 7. Discussion

The importance of neutrophils and their involvement in the immune response has been evident since 1884 when the phenomenon of phagocytosis was first described. These cells, the most abundant of leukocytes, have a short life span and have the role of monitoring infectious and inflammatory processes, with them being the main effectors of the innate immune system. Their interaction with other types of leukocytes, modulating their immune role, explains the choice of association of these ratios as possible biomarkers in autoimmune diseases, cardiovascular diseases, infections, and most types of cancer [31].

Multiple factors such as age, stress, and diseases such as diabetes and coronary heart disease and conditions such as anemia and stroke can influence NLR, with a value between 1 and 2 being considered normal. A so-called “gray zone” between 2.3–3 can serve as an early warning of some pathological conditions including psychiatric disorders, cancer, and atherosclerosis, with them being sensitive to infections, inflammation, and sepsis. Related not only to the cancer-specific mortality and overall cancer mortality of patients but also associated with the response to immunotherapy, NLR is promoted as a cheap and accessible biomarker. A value of three is generally recommended as a cut-off value for prognosis and the decrease in NLR values below seven is generally associated with a reduction in the risk of mortality in severe conditions. Zahorec considers NLR as a future biomarker related to cellular immune activation, an index of stress and inflammation [32].

A meta-analysis involving 14 studies and 5274 patients evaluated the value of NLR, PLR, and MLR as prognostic markers in endometrial cancer, starting from the premise that until now conclusive positive results have only been obtained in cervical cancer, breast cancer, ovarian cancer, and other types of solid tumors. NLR or PLR were correlated with disease-free survival (DFS), with increased values being associated with an unfavorable prognosis in endometrial cancer, but MLR was not associated with OS or DFS [33]. MLR, NLR, PLR, and D-dimer were correlated with clinical outcomes in lung cancer patients treated with surgery. For all these variables, lower values were identified as being associated with better OS and PFS. In the multivariate analysis, the lower MLR value was an independent biomarker for better OS and PFS [34]. By analyzing the results using Cox regression, the prognostic value of NLR and PLR on a set of 1435 cases from the University of Malaya Medical Center Breast Cancer Registry, Koh and colleagues identified both NLR and PLR as prognostic factors in breast cancer. The authors consider that additional studies are needed because this biomarker obviously brings added value if it is included in the prognostic models for breast cancer [35]. A retrospective analysis including 152 prostate cancer patients treated with radiotherapy at the Department of Radiotherapy at the Maria Sklodowska-Curie National Institute of Oncological Research (Gliwice, Poland) aimed to evaluate the prognostic value of the pretreatment values of not only PLR, NLR, lymphocyte-to-monocyte ratio (LMR), and platelets (PLT) but also other biological laboratory values including red blood cells (RBCs), prostate-specific antigen (PSA) level, Gleason score, and factors related to the patient including age, smoking status, and comorbidities. NLR, PLR, leukocyte count, and pre-treatment RBC were identified as independent prognostic factors [36,37].

Even if in high-income countries, especially those in Western Europe and North America, human papillomavirus (HPV) involvement in the etiology of HNSCC is already common; the studies proposed in Romania by Ursu et al. demonstrate that the vast majority of analyzed cases are not related to HPV infection. In 26 cases evaluated for oncogenic viruses’ involvement in HNSCC etiology, 23 of the cases are related to at least one of the viruses, but it should be mentioned that no case was associated with the alpha or beta types of HPV infection. Only 23 cases out of 189, representing 12.2% of the total, were identified as HPV DNA-positive in another study conducted by Ursu and colleagues in northeastern Romania, with half of the cases being oropharyngeal cancers. The authors mention that only a small subset of HNSCC cases were associated with HPV. Even when taking these data into account, the lack of fully reimbursed standard evaluation of HPV status in our country, or at least p16 from immunohistochemistry as a surrogate marker, is obviously a source of uncertainty and possible TNM staging errors for oropharyngeal cancers [38,39].

The limits of the study must also be mentioned, including the uncertainty induced by the presence of comorbidities in some cases that have been shown to be associated with the two facets of the immune system (the innate immune response and lymphocyte-mediated adaptive immunity), with them being correlated with other diseases [40,41]. Also, the variability in the chemotherapy sequences and protocols of radiotherapy and chemotherapy and the lack of complete data regarding smoker status must also be mentioned as sources of uncertainty.

Even if the study confirms the prognostic value of NLR, PLR, and MLR, the weak correlation with OS in the entire patient lot, associated with evidence of a difference in median OS for using the reference interval proposed by the meta-analysis by Cho et al., advocates the use of the concept of the reference interval and not the cut-off value [14]. By being cheap and accessible, these biomarkers could also be evaluated in departments with limited resources; their implementation in prognostic scores also brings economic advantages in limiting the costs associated with the management of advanced stages of HNC. The particularities of the subtype associated with HPV and the tendency to not only de-escalate treatment to improve quality of life but also to escalate therapy for cases resistant to chemotherapy and radiotherapy justify the interest in identifying new biomarkers. Considering multiple confounding factors, including any modulators of inflammation and immune response, it is necessary to not only refine the criteria for the inclusion of HNC patients in prospective studies but also to focus attention on the evaluation of MLR and PLR, which are less often investigated in studies than NLR. With certainty, in an era of immunotherapy, the prospects of including immune checkpoint inhibitors (ICIs) even in locally advanced disease will offer new values to these biomarkers due to the physiopathological mechanisms involved.

## 8. Conclusions

The comparative analysis of the data in the group with NLR < 2 and NLR ≥ 6 highlights an advantage of 4.4 months in median OS in favor of the group with low values of NLR. Not only the role of borderline NLR values (between 2 and 6) as prognostic markers in HNSCC but also the inclusion of PLR and MLR in a prognostic score must also be defined in the future. Prospective studies with more uniformly selected inclusion criteria could demonstrate the value of pre-treatment NLR, PLR, and MLR for treatment stratification through the intensification or de-escalation of non-surgical curative treatment in HNSCC.

## Figures and Tables

**Figure 1 diagnostics-13-03396-f001:**
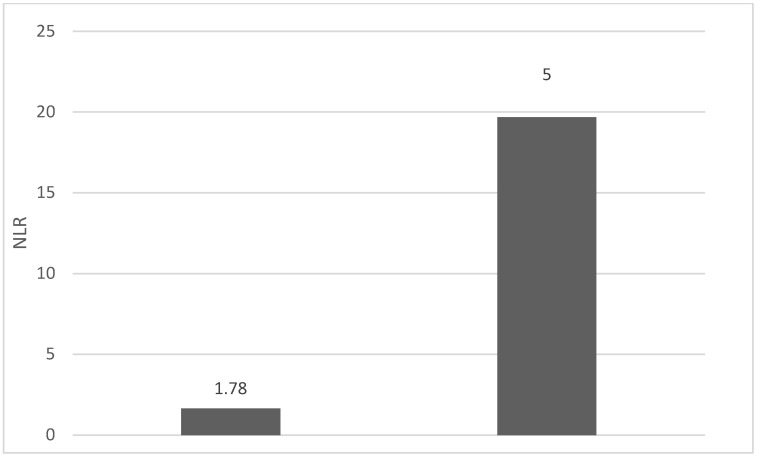
Mean NLR for patients with NLR values between 0 and 2 (**left**) and higher than 6 (**right**).

**Figure 2 diagnostics-13-03396-f002:**
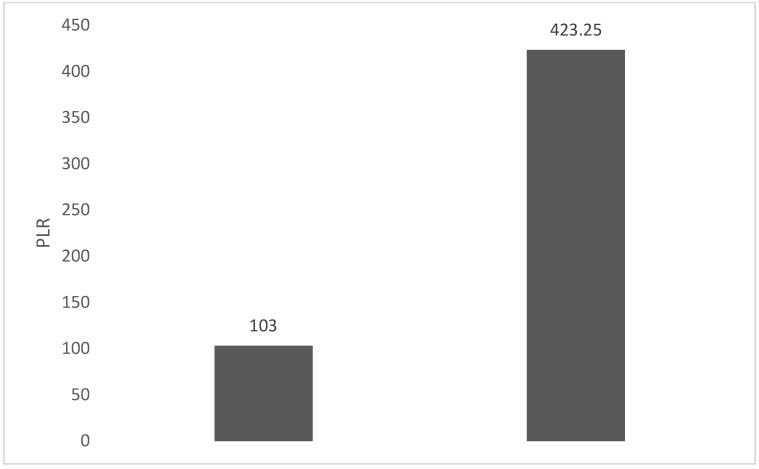
Mean PLR for patients with NLR values between 0 and 2 (**left**) and higher than 6 (**right**).

**Figure 3 diagnostics-13-03396-f003:**
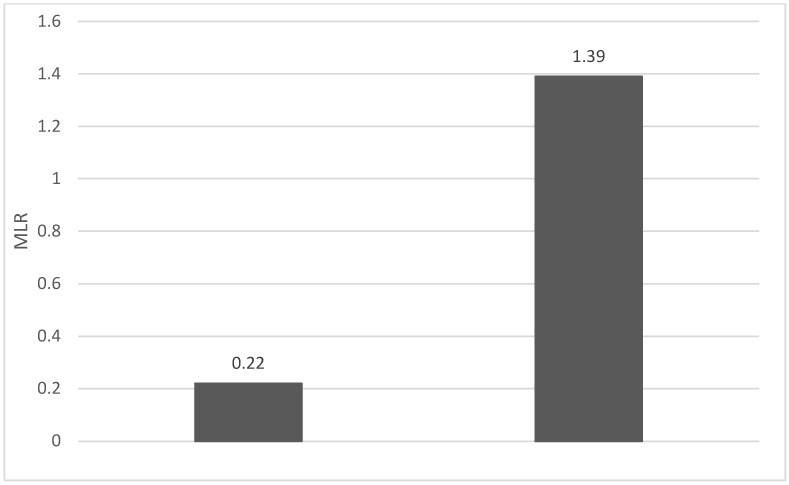
Mean MLR for patients with NLR values between 0 and 2 (**left**) and higher than 6 (**right**).

**Figure 4 diagnostics-13-03396-f004:**
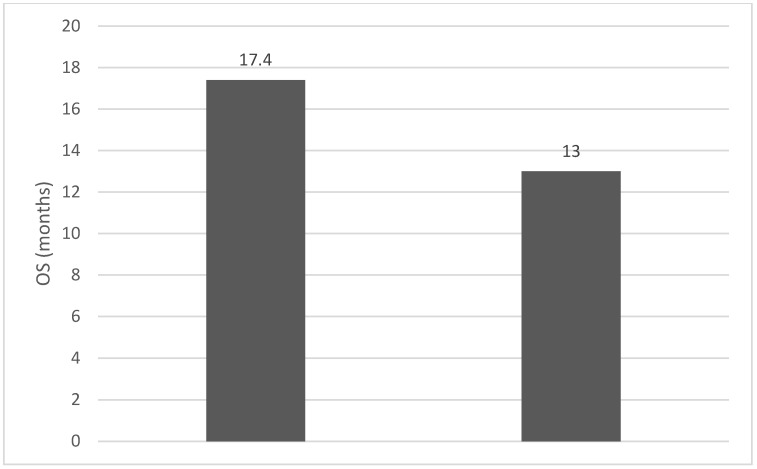
Median OS (months) for patients with NLR values between 0 and 2 (**left**) and higher than 6 (**right**).

**Figure 5 diagnostics-13-03396-f005:**
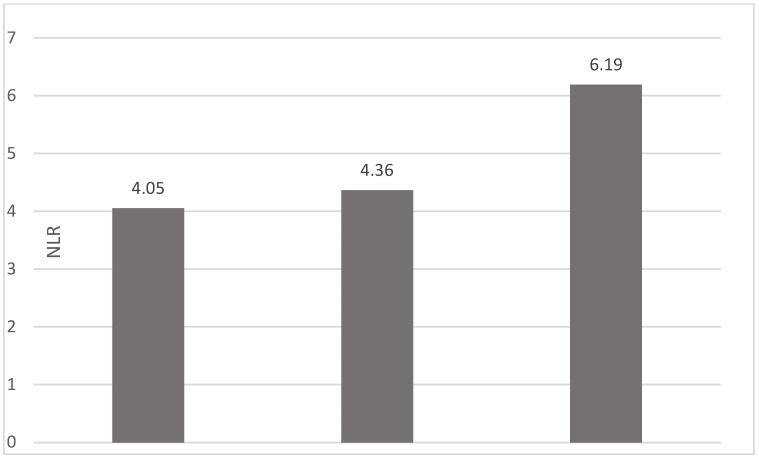
Median NLR nadir values for patients with laryngeal, oropharyngeal, and oral cavity HNSCC (from **left** to **right**).

**Figure 6 diagnostics-13-03396-f006:**
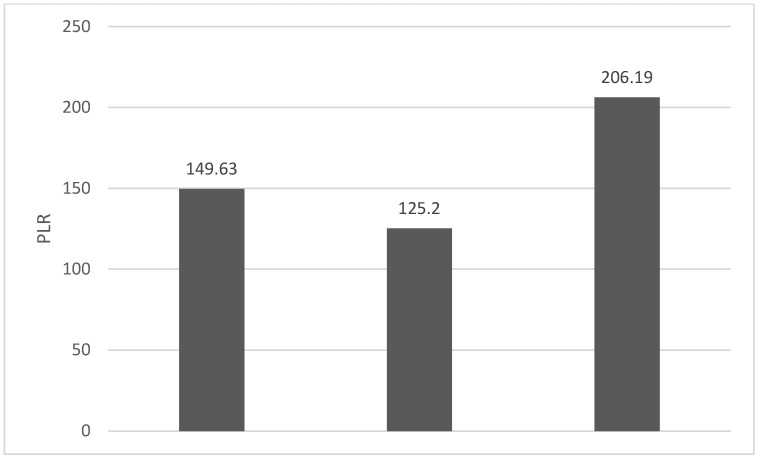
Median PLR nadir values for patients with laryngeal, oropharyngeal, and oral cavity HNSCC (from **left** to **right**).

**Figure 7 diagnostics-13-03396-f007:**
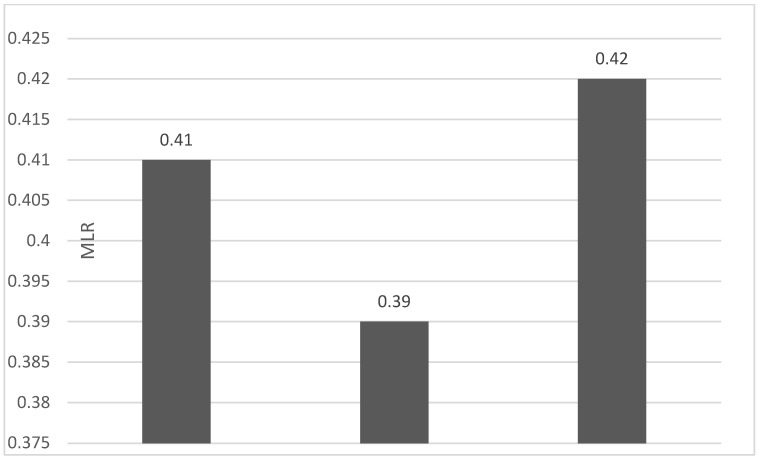
Median MLR nadir values for patients with laryngeal, oropharyngeal, and oral cavity HNSCC (from **left** to **right**).

**Table 1 diagnostics-13-03396-t001:** NLR, PLR, and MLR, predictive and prognostic biomarkers in HNC.

Investigated Ratio(s) as Biomarker(s)	Anatomical Site of Cancer/Histology	Number of Case	Results/Conclusion	Cut-Off Value(s)	Reference
NLR, PLR, LMR, and monocyte-to-white blood cell ratio (MWR)	Laryngeal carcinoma	50	NLR, PLR, LMR, and MWR could be used as prognostic and diagnostic markers in laryngeal cancer; their combination increases the accuracy of the prediction.		Li et al., 2022 [9]
NLR and PLR	Not specified	170 histological confirmed cases + 80 case in the control group	Increased NLR and PLR values are correlated with poor prognosis		Seetohul et al., 2019 [10]
NLR and PLR	Laryngeal squamous cell carcinoma	290	Preoperative NLR and PLR can be used as prognostic markers; their accuracy increases if they are used in combination	2.22 for NLR and 114 for PLR	Tu et al., 2018 [11]
NLR	Nasopharyngeal carcinoma	190	A high NLR was identified as a poor prognostic factor for nasopharyngeal cancer in Taiwan	3.6	Liao et al., 2018 [12]
NLR, PLR, MLR, and systemic immune inflammation (SII) index	HPV-related and HPV-unrelated oropharyngeal cancer	127	The studied immune ratios could be stratification factors in both HPV− and HPV+ cases	NLR > 2.13, SII > 448 for OS and NLR > 2.29, SII > 462.58 for DFS	Brewczyński et al., 2021 [13]
NLR	HNSCC	25 studies in 24 articles and 1536 cases	Pretreatment NLR values below 2 and above 6 could be more conclusive biomarkers of prognosis	NLR < 2, 2 to 6, and ≥6	Cho et al., 2018 [14]
NLR and PLR	HNSCC	28 cohorts involving 6847 cases	High pretreatment NLR predicted poor OS, DFS, and cancer-specific survival. PLR was not associated with OS or DFS		Yang et al., 2019 [15]
NLR, MLR, PLR, alkaline phosphatase (ALP), and l actate dehydrogenase (LDH)	Laryngeal squamous cell carcinoma	361	Elevated preoperative NLR, PLR, MLR, and ALP are predictors of worse survival; NLR and postoperative MLR were identified as independent prognostic markers		Chen et al., 2018 [16]
NLR	Nasopharyngeal carcinoma	463	A value of NLR = 3 is an independent poor prognostic factor	3	Setakornnukul et al., 2021 [17]
NLR and PLR	Head and neck cancer patients treated with (chemo-)radiation	186	A higher NLR is associated with OS but not associated with recurrence-free survival (LRFS), distant recurrence-free survival (DRFS), and acute toxicity grade ≥ 2; PLR was not correlated with outcome or toxicity		Bojaxhiu et al., 2016 [18]
NLR	HNSCC	3770	Elevated NLR predicts worse outcomes		Takenaka et al., 2017 [19]
Platelet count and PLR	HNSCC	8 studies including 4096 patients and 9 studies including 2327 patients	Elevated platelet count and PLR are associated with poor prognosis		Takenaka et al., 2018 [20]
NLR	HNSCC	14 studies involving 929	NLR predicts treatment results in immune checkpoint inhibitors (ICIs)		Takenaka et al., 2022 [21]
NLR	HNSCC	24 articles and 6479 cases	An elevated NLR is a predictor of a poor OS		Mascarella et al., 2018 [22]
NLR and PLR	HNSCC	273	PLR and NLR are independent predictors of mortality and recurrence, respectively		Rassouli et al., 2015 [23]
NLR and PLR	HNSCC	156	An NLR higher than the threshold is associated with an unfavorable evolution. NLR is an independent predictor of five-year overall survival. Neither PLR nor NLR are correlated with tumor recurrence	NLR = 3.9	Szilasi et al., 2020 [24]
Lymphocyte-to-monocyte ratio (LMR)	HNSCC	4260	An elevated LMR may be a predictor of favorable prognosis		Tham et al., 2018 [25]
LMR	HNC	1431	Dynamic delta-LMR during radiotherapy is a simple and inexpensive marker for freedom from metastasis and OS		Lin et al. 2020 [26]
NLR and PLR	Paranasal sinus	215	NLR and PLR are independent prognostic factors of DFS. Higher pretreatment NLR and PLR are related to poor prognosis	NLR = 2.6; PLR = 156.9	Turri-Zanoni et al., 2017 [27]
Fibrinogen (F) and NLR, F-NLR score	Hypopharyngeal carcinoma	111	F-NLR score could stratify patients into prognostic groups		Kuwahara et al., 2018 [28]
NLR and MLR	Early-stage (T1–T2) oral squamous cell carcinoma (OSCC) of the tongue	102	NLR and MLR independent predictors of OS	NLR = 2.96	Ventura et al., 2021 [29]
NLR and PLR	Laryngeal cancer	5716 patients from 20 studies	A higher NLR predicts poor PFS and OS and a higher PLR predicts poor OS		Hu et al., 2022 [30]
NLR, PLR, and NLR/PLR relationship	Laryngeal carcinoma	5716 patients from 20 studies	NLR is associated with poor OS, PFS, and DFS; a higher PLR is a marker of poor OS		Hu et al., 2022 [30]

**Table 2 diagnostics-13-03396-t002:** Patients’ characteristics.

Characteristics	N (Total = 39)	%
Age at the time of diagnosis		
Median (range)	64.84 years (48–86 years).	-
Histology		
squamous cell carcinomas (SCC)	39	100
Anatomical tumor site		
oropharynx	11	28.2
oral cavity	13	33.33
larynx	7	17.94
hypopharynx	2	5.1
nasopharynx	1	2.55
unknown primary	2	5.1
sinonasal	1	2.55
parathyroid	1	2.55
NLR		
Mean (range)	6.22 (1.24–69)	
PLR		
Mean (range)	203.17 (61.3–1775.0)	
MLR		
Mean (range)	0.53 (0.12–5.5)	
Overall survival (OS)		
Median (range)	17.92 (1–120)	

**Table 3 diagnostics-13-03396-t003:** Heatmap including NLR, PLR, MLR, and OS data for the 39 cases included in the study.

NLR	PLR	MLR	OS (Months)
2.07	110.71	0.36	12
2.74	83.69	0.41	6
1.25	74.06	0.14	15
1.24	61.63	0.22	42
4.57	201.55	0.46	4
3.03	127.91	0.39	15
1.83	78.98	0.22	23
4.44	244.14	0.78	3
15.50	139.58	0.76	78
2.51	188.72	0.22	6
2.01	183.14	0.18	6
1.88	88.63	0.14	4
1.48	70.43	0.13	18
8.91	142.34	0.81	13
1.40	88.44	0.18	37
5.04	379.33	0.23	11
2.36	182.23	0.14	9
1.67	89.63	0.12	13
31.85	392.75	0.73	7
4.33	160.91	0.34	1
3.71	107.14	0.43	11
2.56	170.00	0.19	8
8.61	146.50	1.00	11
7.31	327.94	0.84	11
2.75	111.71	0.33	13
1.39	143.59	0.30	17
1.37	201.34	0.46	21
9.22	247.59	0.60	16
2.97	118.70	0.30	3
6.98	214.29	0.85	45
2.69	331.18	0.15	11
69.00	1775.00	5.50	1
1.63	117.50	0.25	3
4.10	155.69	0.45	3
2.78	87.04	0.37	120
4.74	174.14	0.50	69
4.96	131.37	0.56	3
2.34	137.21	0.30	8
3.56	136.92	0.38	17

## Data Availability

Not applicable.

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
