# Peer review of "The Prognostic Value of Platelet–Lymphocyte Ratio, Neutrophil–Lymphocyte Ratio, and Monocyte–Lymphocyte Ratio in Head and Neck Squamous Cell Carcinoma (HNSCC)—A Retrospective Single Center Study and a Literature Review"

_diagnostics, 2023, doi:10.3390/diagnostics13223396_

Round 1
Reviewer 1 Report (Previous Reviewer 2)
Comments and Suggestions for Authors
hello
the paper was upgraded and improved since it last time was last in the review process
abstract is more structurised but should be at least 150words, in current for is too long
the introduction section is too long
does all of those information in the introduction needs to be there?
results are written in half of a page - please improve them, devide into section/subsections, discuss patients age, gender, type of HNC cancer etc - to improve them
all of the graphics should be explained in the results chapter
I strongly advice the Authors too look as a comparison, other papers from mdpi/diagnostics
please improve the paper, since its interesting but recquires some work
Author Response
Dear Reviewer 1,
We thank you for the relevant recommendations and the time allocated for the evaluation of the manuscript. First of all, we reduced the dimensions of the abstract. The introduction has been reduced and distinct sub-chapters have been proposed that we considered necessary to clarify different aspects. However, considering the great weight of these theoretical paragraphs, we proposed adding ("and a review from the literature") in the title. The results section has been substantially expanded with new data from the study and with 3 more graphs analyzing a comparison for the most frequent subtypes of HNSCC encountered in the study group. Also, the conclusions were reduced and some ideas were moved to the discussion section. All proposed changes have now been marked in yellow.
We hope you will appreciate this new version of the manuscript.
Kind Regards,
Camil Mirestean
Reviewer 2 Report (Previous Reviewer 1)
Comments and Suggestions for Authors
I still see the same problems that I mentioned before.
long discussion
Still using "your" instead of "our". like in the results "the relationship between your variables was only weak"
Other than that the manuscript is improved greatly.
minor revision as mentioned in the comments.
Author Response
Dear Reviewer 2,
We thank you for the relevant recommendations and the time allocated for the evaluation of the manuscript. First of all, at the suggestion of reviewer 1, we reduced the dimensions of the abstract. The introduction has been reduced and distinct sub-chapters have been proposed that we considered necessary to clarify different aspects. However, considering the great weight of these theoretical paragraphs and discussions, we proposed adding ("and a review from the literature") in the title. However, the results section was substantially expanded with new data from the study and with 3 more graphs analyzing a comparison for the most frequent subtypes of HNSCC encountered in the study group. Also, the conclusions were reduced and some ideas were moved to the discussion section. All proposed changes have been marked in yellow.
We hope you will appreciate this new version of the manuscript.
Kind Regards,
Camil Mirestean
Round 2
Reviewer 1 Report (Previous Reviewer 2)
Comments and Suggestions for Authors
hello
dear authors
paper looks better and its improved
the introduction is still confusing but its better
I think it should be handled tp main editor to check if it fits the criteria
so far, its an interesting paper, discussing some missing features on HNSCC
thank you
This manuscript is a resubmission of an earlier submission. The following is a list of the peer review reports and author responses from that submission.
Round 1
Reviewer 1 Report
Comments and Suggestions for Authors
Thank you for the effort which indeed improved the manuscript. However, I still think that the introduction is too long (almost 5 pages)
The weak correlation between your data doesn't support your conclusion in my opinion.
You still say "your results" instead of "our results" throughout the manuscript
Reviewer 2 Report
Comments and Suggestions for Authors
hello
interesting paper
the abstract is too long and not well organised
introduction OK
paragraph 1.1 should be either placed in -partly in the introduction, 2nd part in the material and methods section to clearly describe the purpose and methods used in the following study
The blue highlighted text lines in the paper manuscript are confusing for me - please remove and change it
minor mistakes with citations and names of authors - like; Brewczynski et al etc- should be corrected
material and methods should be more structurised
results section should include the results of each blood marker, cancer/tumor type and other correlation that could more briefly describe the results
results are too short and not descriptive, without any new insights
in discussion - please write, how the following authors findings and results might improve patients clinical care in oncology wards?
describe top 5 highlighted features of this paper
Reviewer 3 Report
Comments and Suggestions for Authors
since the authors have made the requested corrections, I have no further comments.